# New Way of Synthesis of Few-Layer Graphene Nanosheets by the Self Propagating High-Temperature Synthesis Method from Biopolymers

**DOI:** 10.3390/nano12040657

**Published:** 2022-02-16

**Authors:** Alexander Voznyakovskii, Aleksey Vozniakovskii, Sergey Kidalov

**Affiliations:** 1Institute for Synthetic Rubber, 198035 Saint-Petersburg, Russia; voznap@mail.ru; 2Ioffe Institute, 194021 Saint-Petersburg, Russia; alexey_inform@mail.ru

**Keywords:** graphene, self-propagating high-temperature synthesis, few-layer graphene, biopolymers, starch, lignin, tree bark, carbonization of biopolymers, SHS, FLG

## Abstract

For the first time, few-layer graphene (FLG) nanosheets were synthesized by the method of self-propagating high-temperature synthesis (SHS) from biopolymers (glucose, starch, and cellulose). We suggest that biopolymers and polysaccharides, particularly starch, could be an acceptable source of native cycles for the SHS process. The carbonization of biopolymers under the conditions of the SHS process was chosen as the basic method of synthesis. Under the conditions of the SHS process, chemical reactions proceed according to a specific mechanism of nonisothermal branched-chain processes, which are characterized by the joint action of two fundamentally different process-accelerating factors—avalanche reproduction of active intermediate particles and self-heating. The method of obtaining FLG nanosheets included the thermal destruction of hydrocarbons in a mixture with an oxidizing agent. We used biopolymers as hydrocarbons and ammonium nitrate as an oxidizing agent. Thermal destruction was carried out in SHS mode, heating the mixture in a vessel up to 150–200 °C at a heating speed of 20–30 °C/min and keeping at this temperature for 15–20 min with the discharge of excess gases into the atmosphere. A combination of spectrometric research methods, supplemented by electron microscopy data, has shown that the particles of the carbonated product powder in their morphometric and physical parameters correspond to FLG nanosheets. An X-ray diffraction analysis of the indicated FLG nanosheets was carried out, which showed the absence of formations with a graphite crystal structure in the final material. The surface morphology was also studied, and the IR absorption features of FLG nanosheets were analyzed. It is shown that the developed SHS method makes it possible to obtain FLG nanosheets with linear dimensions of tens of microns and a thickness of not more than 1–5 graphene layers (several graphene layers).

## 1. Introduction

Graphene materials such as graphene nanoplates (GnP), graphene nanosheets (GnS), graphene oxide (GO), reduced graphene oxide (rGO), and so forth, are successfully used by researchers around the world in various promising areas. There are the following areas: as an additive in the polymer [1] and metal composites [2]; as an effective sorbent of radionuclides [3]; as a material for creating supercapacitors [4], and so forth. However, despite obtaining interesting results using graphene materials, industrial implementation has not yet occurred due to the imperfection of the methods of their synthesis.

Chemical vapor deposition [5] and epitaxial growth on single-crystal SiC [6] make it possible to obtain graphene sheets with few defects in the form of films. However, these methods require expensive equipment and do not provide the high performance needed to synthesize graphene structures for application in industry. 

The classical methods for obtaining GO are the Brodie method (proposed in 1859 by Benjamin Brodie) [7] and the Hammers method [8], which have been significantly improved by now. These methods are based on the exfoliation of graphite with concentrated acids followed by oxidation. As a result, a three-dimensional structure of graphite oxide is formed, which is then exfoliated to form GO nanosheets [9]. In the oxidation process, in addition to the formation of oxygen-containing groups, defects in graphene oxide layers are also formed, which are defects in the carbon sublattice with the absence of two or more carbon atoms [10]. rGO can be obtained by both physical and chemical reduction of graphene oxide [11]. In addition, the occurrence of excess energy and harmful chemicals on the initial formation of graphite leads to significant structural defects in the planes of hexagons of graphene sheets (the presence of vacancy defects, substitution atoms, etc.).

Another approach to the chemical exfoliation of graphite involves the intercalation of graphite with alkali metals such as potassium and lithium, followed by the exfoliation of such intercalated graphite compounds in several layers of graphene nanosheets [12,13].

This approach is much more straightforward and more economical than the previous one. However, this is time-consuming, and hazardous strong acids are used throughout the entire process. This factor also leads to a decrease in the quality of graphene nanosheets, an increase in the cost, environmental problems, and a reduction in the productivity of the method.

The complexity, environmental unsafety, and low quality of the synthesized graphene structures (due to defects in their cellular structure) of the considered methods make them unpromising for obtaining graphene structures in volumes exceeding the volumes required for interlaboratory studies. As a result, many research groups began looking for alternative methods for synthesizing graphene structures.

For example, several authors used a well-practiced hydrothermal synthesis method to obtain graphene structures.

Reference [14] reported the hydrothermal synthesis of GO particles using glucose as a precursor. The hydrothermal method for the synthesis of nanodispersed particles is fairly simple hardware, and a safe, controllable, environmentally friendly, and inexpensive method. Accordingly, the complexity of these parameters can provide great potential for the mass production of high-quality GO. In a similar work [15], rGO was obtained by hydrothermal synthesis using glucose as a precursor. The resulting rGO was further treated with ammonia, making it possible to synthesize particles of few-layer graphene nanosheets (FLG) in gram within an acceptable time interval.

In some cases, FLG nanosheets may be obtained directly by mechanical activation [16]. In this work, the authors applied the well-known elementary method of sonication of peanut shells (PS) to obtain few-layer nanosheets like FLG nanosheets. The authors report that the nanosheets obtained in this work have an extremely high specific surface area (up to 2070 m^2^/g) and developed microporosity (up to 1.33 cm^3^/g).

In several works, the self-propagating high-temperature synthesis (SHS) process is used to synthesize graphene nanostructures. Physically, SHS is a process of moving a wave of a strongly exothermic reaction through a mixture of reagents (oxidizing agent and reducing agent), in which heat release is localized in the layer and is transferred from layer to layer using heat transfer. Typical characteristics of the SHS process are as follows: flame front propagation speed −(0.1–20) cm/s; maximum combustion temperature −(2300–3800) K; the heating rate of matter in the wave is (10^3^–10^6^) deg/s. [17].

In particular, for the synthesis of graphene and few-layer graphene in [18,19] the classical SHS process was used, using calcium carbonate (CaCO_3_, 99.5%) and magnesium Mg to carry out the SHS reaction. Carbamide (CO(NH_2_)_2_, 99.5%) and carbon dioxide (purity 99.9%) were used as the carbon source. In references [20,21], for the synthesis of few-layer graphene, they used a similar mixture using Mg and standard CaCO_3_. Another work using the advantages of SHS [22] used Magnesium powder (200 mesh, 99.0%), polyvinyl alcohol powder (PVA, Mn = 1750), AR grade carbonate salts (Calcium carbonate, CaCO_3_, basic magnesium carbonate, 3MgCO_3_·Mg(OH)_2_·3H_2_O) and AR grade glucose (99.5%). Few-layer graphene was also obtained.

In this paper, we report on a new technique for synthesizing large volumes of FLG nanosheets based on the nontraditional self-propagating high-temperature synthesis (SHS) method. 

From our point of view, to a certain extent, the SHS method can be considered an analog of detonation synthesis of nanodiamonds [23]. Detonation synthesis implies the occurrence of two sequential processes organized in a certain way: explosive de-composition of, used as a precursor, a mixture of two cyclic organic compounds (trinitrotoluene and cyclotrimethylenetrinitramine) and the subsequent process of self-organization of degradation products in the front of a shock wave. Under the influence of extremely high pressure and temperature, 3D carbon nanostructures—detonation nanodiamonds—are formed under specific conditions of detonation synthesis.

The implementation of detonation synthesis shows no theoretical restrictions on the synthesis of 2D carbon nanostructures in the sequence of processes “destruction-self-organization”. Chemical reactions under the conditions of the SHS process proceed according to a specific mechanism of nonisothermal-branched chain processes, which are characterized by the combined action of two fundamentally different factors accelerating the process: avalanche multiplication of active intermediate particles and self-heating. In contrast to thermal ignition, branched-chain ignition is caused by avalanche multiplication of active intermediate products—free atoms, radicals and sometimes-excited particles—in their rapid reactions with the initial reagents and among themselves [24].

When choosing a carbonization technique, we proceeded from the fact that the advantage of using the SHS process over the pyrolysis and hydrothermal carbonization processes traditionally used for carbonization of biopolymers [17] is the simplicity of the hardware design of the method. In addition, there are high synthesis rates, the ability to carry out synthesis without a constant supply of energy from external power sources, the ability to carry out synthesis in any atmosphere or a vacuum, the absence of fundamental large-scale limitations [25,26].

We assumed that biopolymers with a cyclic macrochain structure could be an acceptable source of native cycles (pre-cursors) for the SHS process based on the analogy with detonation synthesis.

In formulating this assumption, we relied on comparing graphene to a specific macromolecule, the native element of which (monomer) can be a cycle of six carbon atoms—hexagon. An acceptable source of hexagons can be plant polymers, the macromolecules of which are either formed by cyclic structures (starch, cellulose). The choice of the source of such cycles, as well as the development of a technique for initiating their “bottom-up” self-organization processes, would make it possible to obtain 2D carbon nanostructures on a scale required for real application.

## 2. Preparation and Carrying Out the Process of the SHS Process

Biopolymers carbonization was carried out under the conditions of the process of the SHS method. The process was carried out using a laboratory reactor, a quartz vessel (1 L capacity) with a heating element in the lower part, which provides the initial heating of the reaction zone to the temperature required to initiate the process (220 °C). Heating to a lower temperature did not lead to the start of the synthesis process. Temperature control in the reaction zone was carried out using a thermocouple. The preparation of the synthesis included a number of sequential stages. The pre-dried biopolymers (until weight loss stops) were transferred to a ball mill and processed for 15 min. The average particle size of biopolymers was no more than 100 μm. Next, a mechanical mixture of an oxidizing agent (ammonium nitrate) and a precursor (biopolymers) was prepared in a weight ratio of 1:1. This ratio was experimentally selected as giving the maximum yield (30–35 wt.% according to the initial biopolymer) of the product. The prepared mechanical mixture of the oxidizing agent and the precursor was placed in a “drunken barrel” mixer and processed for 15 min. Then the resulting crushed mixture was transferred into a reactor preheated and purged with a dry argon flow. The beginning and end of the reaction were judged by the beginning and end of the evolution of gaseous reaction products. The duration of the process is 5–8 min. The rest of the reaction mass passes into the gas phase and is captured in a trap cooled with liquid nitrogen. The process flow diagram is shown in Figure 1.

The methodology for producing FLG nanosheets from wood waste includes several steps, as shown in Figure 1.

All raw materials were crushed (grain crusher Kolos 2M, Extego, Moscow, Russia) and dried to constant weight in an ShS-40-02 drying oven at 80 °C. The dried powder was finely ground in a laboratory planetary mill LP-1-HT Machinery and sifted through a sieve to a particle size of the order of 100 μm (±10 μm).

After completing these elementary cleaning and drying steps, the waste and ammonium nitrate (average particle diameter 10 µm) were thoroughly mixed using a “drunken barrel” mixer in a specific ratio.

The mixture of the initial components prepared in this way was slowly heated to a temperature of about 150–200 °C in an oil bath in an air atmosphere before the start of the decomposition of ammonium nitrate.

A schematic representation of the setup for SHS and the SHS process scheme is shown in Figure 2.

The main feature of carbonization in the process of rapid pyrolysis (a technique usually used for the carbonization of plant polymers) is that fast pyrolysis is carried out during the thermal decomposition of woody biomass in the absence of an oxidizing agent. 

The fundamental difference between the carbonization of plant polymers (mainly woody biomass and products of its processing) under the conditions of the SHS process is that in our case, there are active oxidants—nitrogen and oxygen obtained during the decomposition of ammonium nitrate. This is the most crucial differences between carbonization and the SHS method.

Another significant difference between the processes of fast pyrolysis and the SHS process we use is that the carbonization process under conditions of fast pyrolysis is carried out with a constant supply of thermal energy, that is, the process is self-accelerating [27], while the SHS process is self-inhibiting [17].

Based on these two differences in the considered carbonization processes, it was assumed that, under the conditions of the SHS process, the carbonization of plant biomass would proceed only to a solid carbonized product. This moment of the SHS process we use creates the basis for the nucleation of FLG nanosheets with the secession of all gas oxides and metal oxides.

After the completion of the SHS process, a black powder was obtained. The black powder collected in this way at the end of the SHS process was washed with distilled water to remove water-soluble products (oxides, metal nitrates, etc.). The resulting FLG nanosheets were either dried or left in an aqueous suspension. Experimental conditions are optimized to synthesize FLG nanosheets with up to five layers. 

## 3. Characterization

To confirm the successful synthesis of FLG nanosheets, various spectroscopic research methods have been performed, SEM and TEM images were obtained.

Raman spectroscopy (Confotec NR500, 532 nm, SOL Instruments, Minsk, Belarus) at an ex-citing laser length of 532 nm was used to demonstrate the vibrational properties of FLG nanosheets.

Fourier transform infrared spectroscopy was performed using an INFRALUM FT-08 instrument (Lumex Instruments, Fraserview, BC, Canada), to study functional groups on the surface of FLG nanosheets. X-ray diffraction (XRD) was performed using an XRD-7000 (Cu Kα-radiation, λ = 0.154051 nm, Shimadzu, Kyoto, Japan).

SEM and TEM studies of the morphology and structure of FLG nanosheets were carried out using the TESCAN Mira-3M (Brno, Czech Republic) with Electron Dispersive X-ray Spectroscopy (EDX) device attachment (Oxford instruments X-max, Abingdon, UK) and FEI Tecnai G2 30 S-TWIN (50 kV) microscope. UV-VIS spectra investigated on Shimadzu-IS (Kyoto, Japan). The studies based on the method of the X-ray photoelectron spectroscopy were conducted on the instrument Thermo Scientific K-alpha (Waltham, MA, USA).

## 4. Results

To evaluate the success of the above experimental strategy for synthesizing FLG nanosheets using the SHS method, we used various methods to validate the synthesis of FLG nanosheets.

### 4.1. SEM and TEM Studies

The material synthesized by the SHS method was a black, highly dispersed, highly volatile powder. The structure of synthesized FLG nanosheets obtained using SEM and TEM is shown in Figure 3.

As shown in Figure 3, the FLG nanosheets synthesized by the SHS method have linear dimensions of about 10–30 µm. It should be noted that the obtained FLG nanosheets are not strictly plane-parallel and have curvatures. It can be seen from Figure 3a that the samples have a small thickness (possibly a single layer), which makes it possible to distinguish particles lying behind a sheet of few-layer graphene even at low accelerating voltages (20 kV). It should be noted that the cellulose sample is more prone to aggregate formation than the glucose and starch sample.

As shown in Figure 3d–f, the synthesized particles have a few-layer structure formed by the superposition of differently oriented graphene layers. It is also seen that the number of layers in the samples does not exceed 5.

### 4.2. EDX Research

The EDX studies (Figure 4) showed the high purity of the FLG nanosheets obtained by the SHS method.

From the above data on the study of a sample obtained from glucose, it can be concluded that there are no impurities of metals or other elements. The main impurities are oxygen and nitrogen (due to the presence of an oxidizing agent—ammonium nitrate in the SHS process) contained in an amount of no more than 10 and 15 at.%, respectively. Oxygen content in graphene sample up to 15 wt.% is typical for all graphene nanostructures, except graphene oxide, in which the oxygen content can reach 50 wt.%. FLG nanosheets produced from other starting materials had the same impurity purity.

### 4.3. Raman Spectroscopy Data

Raman spectroscopy was performed to confirm the exact synthesis of FLG nanosheets and to assess their quality (Figure 5).

Various active modes are observed in the synthesized samples. The Raman spectra of all samples contain four peaks characteristic of graphite materials, corresponding to D (1340–1345 cm^−^^1^), G (1570–1580 cm^−1^), 2D (2680–2690 cm^−1^) and D+G (2920–2930 cm^−1^).

Similar Raman spectra of graphene nanomaterials (graphene oxide, reduced graphene oxide, graphene nanosheets) were observed in references [28,29,30,31,32].

The G band arises from the planar vibrations of the aromatic carbon sp^2^. The 2D band is a second-order Raman characteristic of crystalline graphite materials and is sensitive to the aromatic C-structure. The D and D+G bands represent Raman features caused by lattice disorder. Disorder in the graphite lattice occurs due to defects associated with vacancies, grain boundaries, carbon particles sp^3^, and the inclusion of O in the aromatic sp^2^ C-network [33].

It was found that comparison of the 2D and D+G Raman bands can indirectly control the electronic structure (π-orbitals). Band D shows partially disordered structures of the sp^2^ carbon atom associated with structural defects, and band G shows the degree of graphitization. The low-intensity ratio of the G band to the D band (I_G_/I_D_ = 0.92, 0.84, and 0.61 for samples synthesized from glucose, starch, and cellulose, respectively) indicates the existence of randomly arranged graphene nanosheets. Based on the data on the position of the 2D peak, the number of layers in the synthesized samples does not exceed 5 [34].

The G-band is Raman-active for the sp^2^-hybridized carbon material, while the D-band is activated only if the defects are involved in double resonance Raman scattering of light near the *K* point of the Brillouin zone. Thus, our results show that the average size of sp^2^ domains changes significantly in samples prepared from lignin relative to samples obtained from cellulose and glucose, since the intensity ratio I_D_/I_G_ is usually used to estimate the size of the sp^2^ domain of graphite-based materials.

In addition to the G and D bands, two weaker Raman bands, called 2D and D+G, are located in the 2920–2930 cm^−1^ region. The 2D band is Raman active for crystal-line graphite materials and is sensitive to the band in the electronic structure of graphite, while the D+G combination mode is caused by the disorder. The I_2D_/I_D+G_ intensity ratios show that the recovery of graphite electronic interfacing for glucose-derived samples is significant.

Thus, it can be concluded that the FLG nanosheets obtained by the SHS method have structural defects and have a slightly curved surface with kinks, breaks, and bends.

### 4.4. X-ray Structural Analysis

An X-ray diffraction analysis was carried out to prove the absence of graphite nanostructures in the obtained material (graphite, thermally expanded graphite, nanographite, etc.), the results of which are shown in Figure 6.

A halo in the range from 10 to 35 degrees and reflections at ~43 degrees strictly proves the absence of any graphite materials, the diffraction patterns of which are characterized primarily by a narrow and intense peak in the region of 26 degrees, as well as a narrow peak with low intensity in the region of 54–55 degrees [35]. As can be judged from the data in Figure 6, the form of the obtained diffraction pattern fully corresponds to the typical curves obtained on 2D carbon structures—graphene nanoplatelets [29].

As seen from Figure 6, a typical diffraction pattern has a wide amorphous halo typical of graphene nanostructures. The absence of any narrow peaks in the diffraction patterns also indicates the absence in our FLG nanosheets of any water-insoluble impurities that could form during SHS process, which is consistent with the EDX data (Figure 4). We also estimated the number of layers in the synthesized FLG nanosheets using the formula *N* = (*L/d*), where *N* is the number of layers; *L* is the thickness of the stack of graphene structures; *d*—interlayer distance [36]. The thickness of the stack of graphene structures was calculated using the Scherrer formula [37]:*L* = *nλ*/*β*⋅cos*θ*,
where *L* is the thickness of the pack; *λ* = 0.154051 nm—radiation wavelength; *θ*—scattering angle; *β*—physical line broadening in the diffractogram in radians (on a 2*θ* scale); *n*—particle shape factor equal to 1. The number of layers in FLG nanosheets obtained by the SHS method from biopolymers is set as a value not exceeding five layers.

### 4.5. FTIR Research Data

FTIR studies were carried out to assess the functional composition of the edges of FLG nanosheets (Figure 7). It is shown that in addition to the typical GnP groups CH, C–OH, CO–C (edge groups), and C=C (the bond between carbon atoms ), the sample contains C≡N groups, the source of nitrogen, which is the oxidizing agent used in the synthesis.

It can be seen that, depending on the initial biopolymer, the content of C≡N, C–O and C–OH groups on the surface of FLG nanosheets changes. Similar results were obtained using lignin as starting material.

### 4.6. Study of the Specific Surface Area

In addition, to characterize the synthesized FLG nanosheets, studies were carried out by the method of gas (helium) pycnometry and the measurement of the specific surface by the BET method, the results of which are presented in Table 1.

As can be seen from the data presented in Table 1, all samples of FLG nanosheets have a developed specific surface area (>500 m^2^/g). Although GNP samples can have a specific surface area of up to 750 m^2^/g [38], usually such high values of the specific surface area are typical for GNP, with a particle size of less than 2 μm [39]. However, in our case, such a developed specific surface area was obtained for FLG nanosheets with linear dimensions of tens of microns (Figure 3). It should be noted that, in contrast to the specific surface area, the true density of the samples does not depend on the type of the initial biopolymer.

Thus, based on the set of data obtained, we can confidently conclude that the structure of the particles obtained in the work corresponds to the structure of the FLG nanosheets.

### 4.7. UV-VIS Research Data

Studies of optical absorption in the UV-VIS range have shown that the spectrum corresponds to graphene structures (Figure 8). The absorption peak lies in the 260 nm range, indicating the practical absence of oxygen on the plane of FLG nanosheets. For glucose, the peak has a maximum at 264 nm, for cellulose at 266 nm, and for starch, it is shifted to 267 nm.

It could be attributed to π–π* transitions of aromatic C–C bonds, which is in agreement with a report for high-quality graphene dispersions by liquid-phase exfoliation and corresponds to FLG nanosheets with up to five layers [40].

### 4.8. XPS Spectra

Figure 9 shows that there are peaks typical for graphene nanostructures: C=C (sp^2^, 284.4 eV) C–OH (285.4 eV), O–C–O/C–OH (286.5 eV), C=O (287.4 eV), OC=O (289 eV), and also the C–N (285.9 eV) peak is observed in the spectrum [41]. The data obtained by the XPS method are in good agreement with the data obtained by the SEM-EDX and FTIR methods. The appearance of the C–N peak is due to using a nitrogen-containing oxidant to synthesize FLG nanosheets by the SHS method.

All spectra demonstrate the presence of the C–N peak in the obtained FLG nanosheets, which fundamentally distinguishes them from GO and rGO, where such peaks are not observed. The intensity of the peaks of the carboxyl and other groups does not exceed the intensity of the C=C (sp^2^, 284.4 eV) peak, which is typical for graphene and is not specific for oxidized forms of graphene nanostructures, primarily graphene oxide [29]. The XPS spectra of samples did not contain elements other than C, O, H and N, indicating the absence of impurities.

## 5. Discussion

We report on an environmentally friendly, reliable, and cost-effective method for mass-producing FLG nanosheets from wood waste.

Raman spectroscopy, TEM, FTIR, UV-Vis, and EDX spectroscopy were performed to confirm the synthesis of FLG nanosheets, including identification of functional groups and quantitative analysis of elements. The proposed SHS method is very effective and allows converting waste from the wood processing industry into FLG nanosheets with a finished product yield of about 40%.

Thus, the results of our investigation showed that the SHS method could be used to obtain FLG nanosheets from polysaccharides (starch, glucose, cellulose, etc.) as precursors. The FLG nanosheets obtained are characterized with linear dimensions of tens of microns and several (up to five) graphene layers thick.

The conducted research allowed us to conclude that the imperfection of the lattice of FLG nanosheets, obtained by carbonization of biopolymers in the process of self-propagating high-temperature synthesis, are free from Stone–Wales defects but are mainly determined by vacancy defects.

The method developed by us allows obtaining large volumes of FLG nanosheets. The use of the developed material in areas requiring large volumes of raw materials and not requiring high-quality single-layer graphene, namely, when creating composite materials, as an effective sorbent and the other regions, will be possible.

Moreover, as our preliminary experiments demonstrate, graphene synthesized by the SHS method can be used for various other applications, such as a material for supercapacitors, fuel cells and a filler for polymer and metal nanocomposites, nanofluids based on water and organic solvents for cooling systems, and so forth.

## 6. Conclusions

For the synthesis of few-layer graphene, the method SHS was applied because it is a versatile method of nanomaterial production. What is new is the use of biopolymers and ammonium nitrate as precursors for the SHS process. Fast self-propagating of reaction in a mixture of ammonium nitrate and biopolymers (glucose, starch, and cellulose) results leads to the effective formation of carbon nanostructures (few-layer graphene) in the purified product. Other biopolymers, such as lignin and tree bark, can be used.

SEM and TEM studies, Raman spectroscopy, X-Ray and UV-Vis spectroscopy, IR spectroscopy and XPS analysis confirmed the graphitic crystalline structure of FLG nanosheets and its thickness of not more than five graphene layers.

The EDX studies showed the high purity of the FLG nanosheets obtained by the SHS method.

This SHS method is environmentally friendly, simple, fast, economical and suitable for the large-scale production of FLG. The preparation of FLG using this technique offers the advantage of simplicity without harmful environmental effects.

Thus, a new high-performance SHS method has been developed to synthesize few-layer graphene nanostructures of a large area from biopolymers, including waste from the woodworking industry—lignin and tree bark. The SHS method will make it possible to obtain graphene nanostructures cheaply in large quantities while solving the environmental problem existing in the world. The SHS method is a breakthrough in the technology and production of graphene nanosheets.

Moreover, this method can solve a significant environmental problem of lignin and tree bark utilization.

The SHS method developed in this article for the production of FLG is promising for creating, for example, doped graphene by choosing various reagents to participate in the SHS reaction.

## Figures and Tables

**Figure 1 nanomaterials-12-00657-f001:**
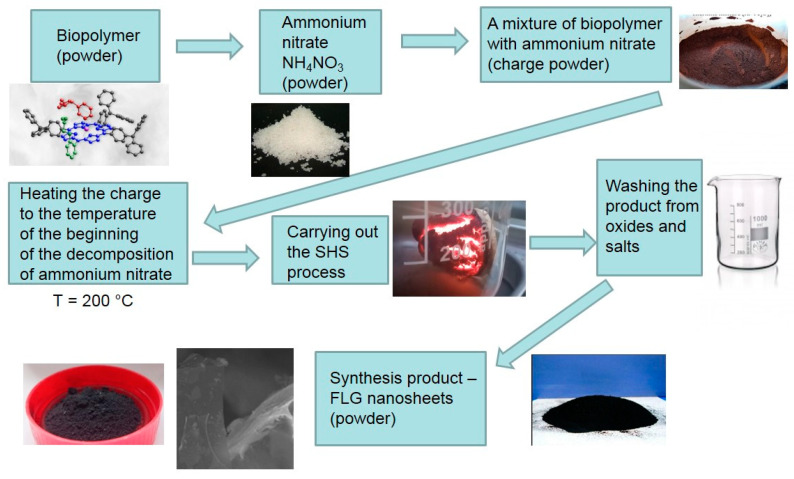
Process flowchart.

**Figure 2 nanomaterials-12-00657-f002:**
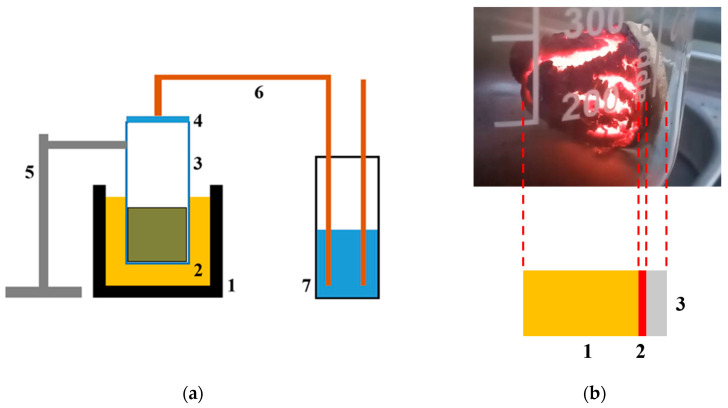
Schematic representation of the setup for SHS (**a**) and the course of the SHS process (**b**). (**a**): 1—oil bath, 2—PVA-300 oil, 3—reactor with charge, 4—reactor cover, 5—tripod-holder, 6—gas exhaust system, 7—water filter. (**b**) The front of the wave of the SHS process is visible, 1—reacted mixture (product), 2—reaction zone (SHS wavefront), 3—unreacted mixture (initial charge).

**Figure 3 nanomaterials-12-00657-f003:**
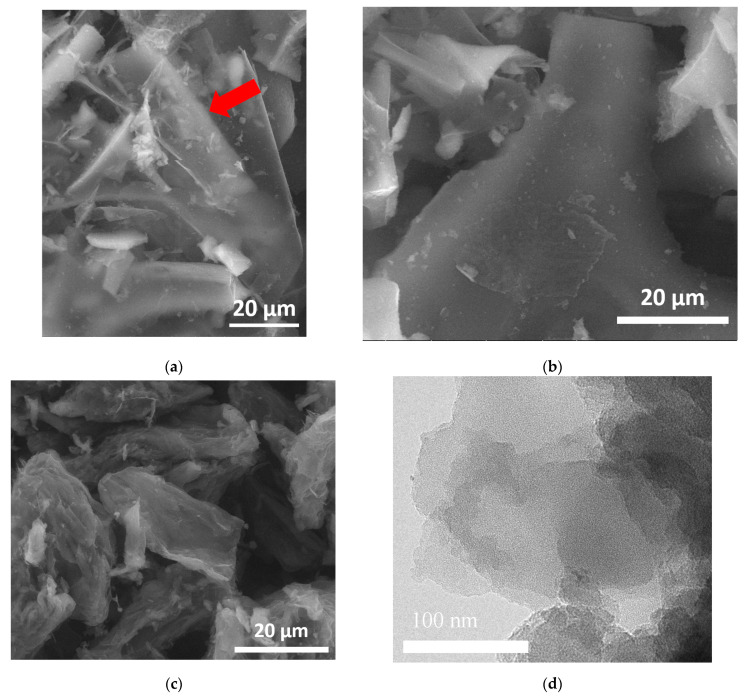
SEM and TEM images of synthesized FLG nanosheets. (**a**–**c**) SEM images—sample synthesized from glucose, starch, and cellulose respectively. Linear scale: a–c—20 µm; (**d**–**f**) TEM images—sample synthesized from glucose, starch, and cellulose respectively. Linear scale: d—100 nm, e—50 nm and f—10 nm. The red arrow marks a particle that is visible through a sheet of few-layer graphene.

**Figure 4 nanomaterials-12-00657-f004:**
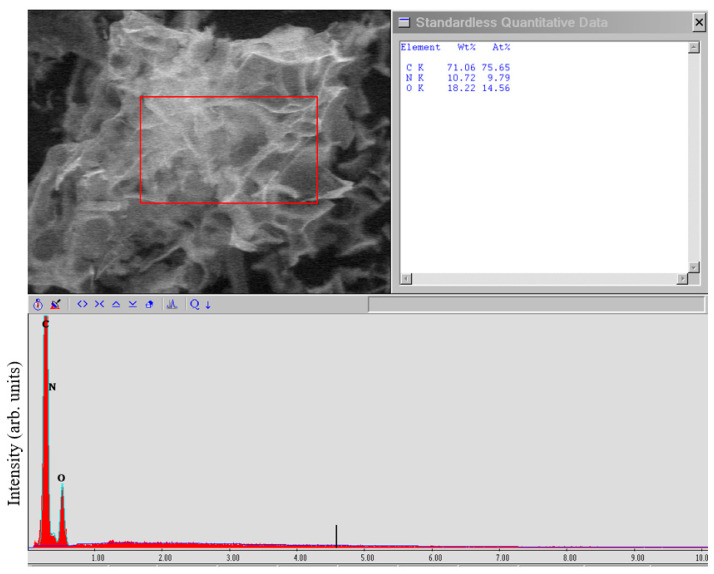
EDX study of synthesized FLG nanosheets from glucose. In bottom picture Y axis—Intensity of EDX signal, arb.units, X axis—electron energy, keV.

**Figure 5 nanomaterials-12-00657-f005:**
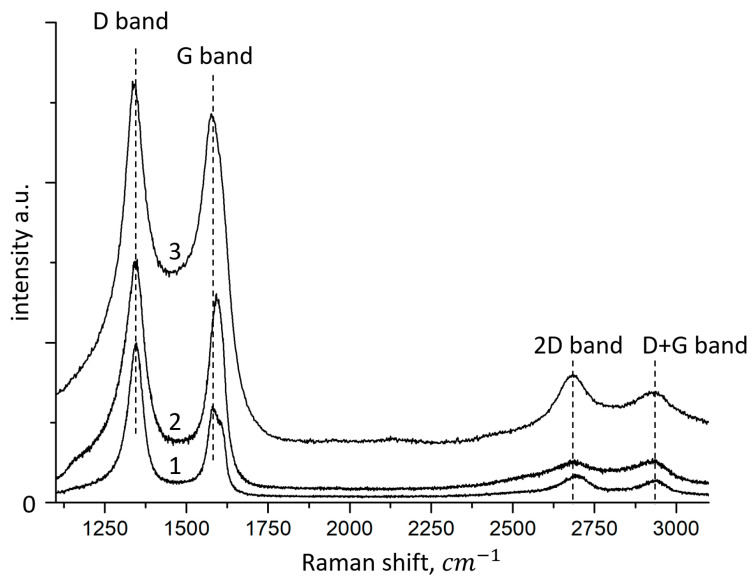
Raman spectra of FLG nanosheets obtained in the SHS process under various conditions and from various precursors. 1—FLG nanosheets derived from glucose, 2—FLG nanosheets derived from cellulose, 3—FLG nanosheets derived from lignin.

**Figure 6 nanomaterials-12-00657-f006:**
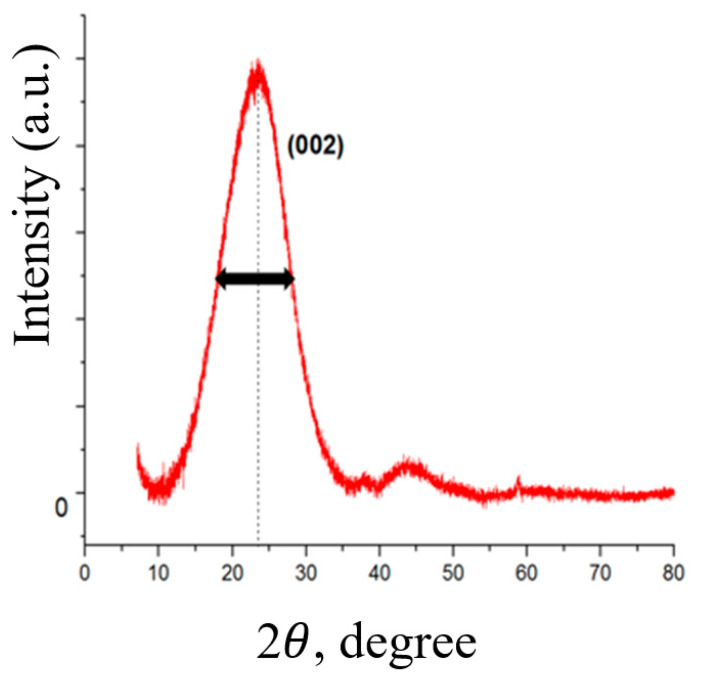
Results of X-ray spectroscopy studies of the synthesized FLG of graphene nanostructures. Typical diffractogram.

**Figure 7 nanomaterials-12-00657-f007:**
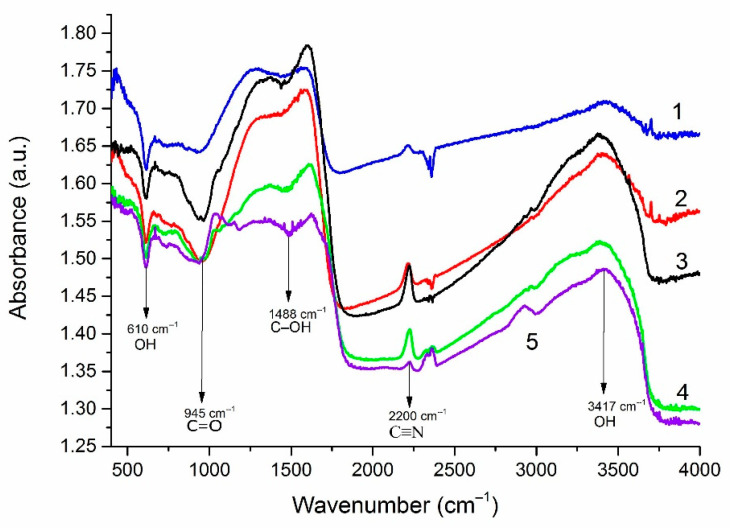
Results of the study of synthesized FLG nanosheets by FTIR spectrometry. The initial charge as a biopolymer includes 1—cellulose, 2—glucose, 3–5—starch. Spectra 3–5 were obtained on FLG nanosheets obtained in a starch-based process under various conditions of the SHS process.

**Figure 8 nanomaterials-12-00657-f008:**
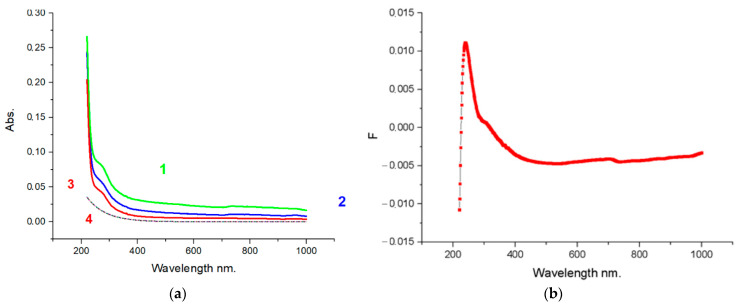
UV-VIS spectra of FLG nanosheets. (**a**) UV-VIS absorption spectra in FLG nanosheets. 1—glucose, 2—starch, 3—cellulose, 4—background spectrum. (**b**) Graph for FLG nanosheets from starch with the background subtracted on residual hydrocarbons that may be present in graphene nanosheets.

**Figure 9 nanomaterials-12-00657-f009:**
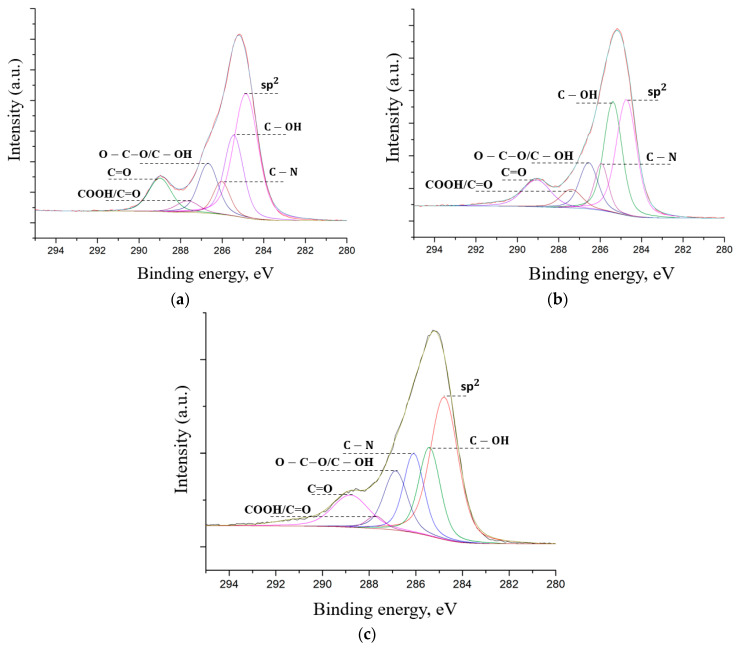
XPS spectra of FLG nanosheets. (**a**) glucose,(**b**) starch, (**c**) cellulose.

**Table 1 nanomaterials-12-00657-t001:** Specific surface area and true density of FLG nanosheets synthesized from various biopolymers.

Sample of FLG	Specific Surface, m^2^/g	True Density, g/cm^3^
From cellulose	672	2.13
From glucose	512	2.11
From Lignin	500	2.12

## Data Availability

Not applicable.

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
