# Peer review of "New Way of Synthesis of Few-Layer Graphene Nanosheets by the Self Propagating High-Temperature Synthesis Method from Biopolymers"

_nanomaterials, 2022, doi:10.3390/nano12040657_

Round 1
Reviewer 1 Report
This paper reports the synthesis of graphene nanosheets using the SHS method. The authors have presented a novel and unique synthesis method. The presented data are not sufficient to support the authors' claim. The manuscript suffers from serious deficiencies and is not suitable for publication in its current form. The following comments need to be addressed to improve the quality of the manuscript.
- Previous works on the SHS process should be included in the introduction section. And the process should be fully explained.
- During the SHS process, what are the temperatures that the mixtures experience? The authors should further elaborate on the SHS process.
- Figure 5: Many previous works on multilayer graphene nanostructures show a sharp and narrow 2D peak, whereas the 2D peak shown here is very broad and low in intensity and similar to those of graphitic material. Can the authors explain why does 2D peak exhibit such a broad profile and contradict what is already established in the literature?
- The quality (resolution) of SEM and TEM images is poor. These images should be replaced with higher-quality images. Furthermore, the TEM images do not support the claim that authors have made regarding the thickness of the Multilayer Graphene Nanosheets. The thickness of up to 5 layers of these graphene-like materials is not demonstrated. Calculating the thickness from the XRD data is not sufficient and should be supported by TEM. The authors should also include a histogram of the layer thickness range.
- Experimental parameter such as particle size of the raw material, the density of the pellet, etc., plays an important role in the SHS process. The authors should explain why only a single experimental condition was used to synthesize the FLG, and on what basis these experimental conditions were chosen?
- The conclusion is too short and is not sufficient. The conclusion section should bring everything together in a logical and systematic manner. It should provide a clear interpretation of the results of your research in a way that stresses the significance of your study.
- This manuscript requires significant editing, as it is not written in sound English.
Reviewer 2 Report
This article reports the synthesis of few-layer graphene nanosheets by the self-propagating high-temperature synthesis method for large-scale production. I recommend publishing after major corrections.
Comment # 1: The title is too lengthy and should be revised to make the manuscript more attractive and attentive for the readers.
Comment #2: Authors should include relevant literature on the synthesis of graphene in the introduction section to emphasize the significance of their work.
Comment # 3: In the introduction, there is no need to write such a big sentence “referred to in the literature as graphene nanoplatelets or GnP” author can simply write “GnP”. Also, the introduction should be more conclusive and precise; it should be explaining recent advancements in the field.
Comment # 4: In the introduction section, the first word starting any sentence must be in capital letters “firstly, the use of hazardous chemicals …”. This type of mistake is so often throughout the manuscript which must be avoided. The experimental part is not written properly, poor sentence structures, the author needs to modify the experimental section.
Comment # 5: The figures in the manuscript can be placed inside a frame; also, figures such as Figures 2, 3, 8, and 9 must be formatted and their captions need revision.
Comment # 6: The author needs to revise the result and discussion section focusing on Raman, XRD, SEM, and XPS data.
Comment #7: The Figure quality is very poor, the authors should revise most of the figures in the manuscript.
Comment #8: The conclusion section must be modified and needs to add highlights of the research article and probable aids that it can lead.
Round 2
Reviewer 2 Report
The introduction part is too long. This should be more conclusive and precise. So authors need to shorten the introduction part. I recommend accepting it for publication after revising the introduction part.
